# FedSGDCOVID: Federated SGD COVID-19 Detection under Local Differential Privacy Using Chest X-ray Images and Symptom Information

**DOI:** 10.3390/s22103728

**Published:** 2022-05-13

**Authors:** Trang-Thi Ho, Khoa-Dang Tran, Yennun Huang

**Affiliations:** Research Center for Information Technology Innovation, Academia Sinica, Taipei 10607, Taiwan; khoatd307@citi.sinica.edu.tw (K.-D.T.); yennunhuang@citi.sinica.edu.tw (Y.H.)

**Keywords:** COVID-19 detection, federated learning, convolutional neural network, differential privacy stochastic gradient descent, spatial pyramid pooling layer, chest X-ray images, COVID-19 symptoms

## Abstract

Coronavirus (COVID-19) has created an unprecedented global crisis because of its detrimental effect on the global economy and health. COVID-19 cases have been rapidly increasing, with no sign of stopping. As a result, test kits and accurate detection models are in short supply. Early identification of COVID-19 patients will help decrease the infection rate. Thus, developing an automatic algorithm that enables the early detection of COVID-19 is essential. Moreover, patient data are sensitive, and they must be protected to prevent malicious attackers from revealing information through model updates and reconstruction. In this study, we presented a higher privacy-preserving federated learning system for COVID-19 detection without sharing data among data owners. First, we constructed a federated learning system using chest X-ray images and symptom information. The purpose is to develop a decentralized model across multiple hospitals without sharing data. We found that adding the spatial pyramid pooling to a 2D convolutional neural network improves the accuracy of chest X-ray images. Second, we explored that the accuracy of federated learning for COVID-19 identification reduces significantly for non-independent and identically distributed (Non-IID) data. We then proposed a strategy to improve the model’s accuracy on Non-IID data by increasing the total number of clients, parallelism (client-fraction), and computation per client. Finally, for our federated learning model, we applied a differential privacy stochastic gradient descent (DP-SGD) to improve the privacy of patient data. We also proposed a strategy to maintain the robustness of federated learning to ensure the security and accuracy of the model.

## 1. Introduction

The severe acute respiratory syndrome coronavirus 2 (SARS-CoV-2) causes coronavirus disease 2019 (COVID-19). It has spread worldwide, resulting in the ongoing 2022 pandemic. With more than 400 million confirmed cases and five million deaths across nearly 223 countries, COVID-19 is continuing to spread around the world, and there is no sign of it stopping. Thus, it has led to a problematic situation for humans in the world until now. Although COVID-19 vaccines have provided an opportunity to slow the spread of the virus and end the pandemic, not enough COVID-19 vaccines are available for everyone in the world to be inoculated until the end of 2024 at the earliest, according to the chief executive of the world’s largest vaccine manufacturer [1]. Moreover, the emergence of COVID-19 virus variants may make the virus more infectious [2] or more capable of causing severe disease [3]. The symptoms of COVID-19 often include fever, chills, dry cough, and systemic pain [4,5]. However, many people are infected with the virus without noticeable symptoms [6,7]. Thus, COVID-19 infection becomes difficult to diagnose. In addition, if the patient is detected early, the disease will be cured more quickly, limiting the spread of the disease. Therefore, it is critical to find a method to assist hospitals in the early diagnosis of COVID-19 patients.

Many researchers have applied artificial intelligence (AI) technology to develop COVID-19 detection models to assist hospitals in detecting patients early. Some of them identified COVID-19 cases based on longitudinal information on patient symptom profiles [8,9,10] and achieved promising results in COVID-19 early detection. Other researchers focused on chest radiography images because most COVID-19 cases display common features on chest radiographs, including early ground-glass opacity and late-stage pulmonary consolidation. It can also be identified through a rounded morphology and a peripheral lung distribution [11,12]. In 2020, research published in the Journal Radiology [13] demonstrated that chest radiography outperformed laboratory testing in detecting coronavirus. Therefore, using chest radiography image analysis can help to screen suspected COVID-19 cases at an early stage. In particular, patient’s symptoms and chest X-ray images have various advantages including high accessibility, affordability, ease of operation, and rapidly prioritizing COVID-19 suspected patients.

Most studies use symptom information [8,9,10], chest X-rays radiography (CXR) [14], chest computed tomography (CT) [15], and lung ultrasound (LUS) [16] as screening methods. These methods heavily rely on shared datasets for the training process. However, based on general data protection regulations [17], patient data privacy must be protected to avoid an attack from malicious attackers because data privacy directly impacts human politics, businesses, security, health, and finances, etc. Therefore, we must find a better way so that machine learning can work collaboratively while maintaining data privacy. One recent method that addresses this problem is federated learning (FL), proposed by Google [18]. Its main idea is to develop a decentralized machine learning model based on datasets from multiple data sources without sharing data. The model updates focus more on the learning task than raw data, and the server only must hold individual updates ephemerally. Therefore, FL offers significant privacy improvements compared to centralizing all training data. Several researchers have applied FL for COVID-19 detection tasks and achieved promising results [19,20,21]. However, some studies have demonstrated that FL may not always provide sufficient privacy guarantees. The sensitive information can still be revealed through model updates [22,23]. For example, Phong [24] demonstrated that the local data information can be revealed from a small portion of gradients, or a possible scenario is that the malicious attacker can reconstruct the training data from gradient information in a few iterations [25].

Unlike existing methods, we did not build a traditional FL system. In this study, we proposed an FL model for COVID-19 detection with higher privacy by adding the differential privacy stochastic gradient descent (DP-SGD) that are resilient to adaptive attacks auxiliary information. We also evaluated the parameters to keep the robustness of FL to ensure the model’s security and accuracy.

In summary, this study makes the following contributions:We proposed a higher privacy-preserving FL model for COVID-19 detection based on symptom information and chest X-ray images collected from multiple sources (that is, hospitals) without sharing data among data owners by adding the differential privacy stochastic gradient descent (DP-SGD) resilient to adaptive attacks auxiliary information;We observed that adding the spatial pyramid pooling (SPP) layer in 2D convolutional neural networks (CNNs) achieve better accuracy on chest X-ray images;We demonstrated that the accuracy of FL for COVID-19 detection reduces significantly for Non-IID data owing to the varying size and distribution of local datasets among different clients. We thoroughly analyzed several design choices (for example, the total number of clients, amount of multi-client parallelism, and computations per client) to improve the model’s accuracy with Non-IID data;We provided a strategy to keep the robustness of our privacy-preserving FL model to ensure the model’s security and accuracy by keeping the fraction of the model constant, scaling up the total number of clients and noise proportionally.

In the remainder of this study, we first review related work in Section 2. We then present our approach in Section 3. Section 4 presents the experimental results, and we finally conclude the study in Section 5.

## 2. Related Works

Many researchers have developed various COVID-19 detection models to help hospitals detect patients early. Most researchers have focused on identifying COVID-19 cases based on chest X-ray and CT images. Horry [14] explored transfer learning for COVID-19 detection using three kinds of medical images (X-ray, ultrasound, and CT scan). Through the comparative study of several popular CNN models, the VGG19 model performed multiple levels of COVID-19 detection for all three lung image models. Afshar presented a capsule framework (COVID-CAPS) to identify COVID-19 cases from chest X-ray images [26]. They demonstrated that the COVID-CAPS outperformed the traditional model. Mukherjee proposed a CNN tailored Deep Neural Network (DNN) algorithm to identify COVID-19 cases using chest X-ray and CXR images [27]. They demonstrated that their model outperformed the other models such as InceptionV3, MobileNet, and ResNet.

Other researchers have used COVID-19 patients’ symptom data. Otoom [28] proposed a real-time COVID-19 detection, treatment, and monitoring system. They used an Internet of Things (IoT) framework to collect real-time symptom data from users, and then identified suspected coronavirus cases and administered appropriate treatment during quarantine. They evaluated the framework’s performance using eight algorithms(support vector machine, neural network, Naïve Bayes, K-Nearest Neighbor, decision table, decision stump, OneR, and ZeroR), five of which achieved an accuracy of more than 90%. Akib Mohi [29] presented a COVID-19 classification system using textual clinical reports. They extracted features using several feature extraction techniques such as bag of words, term frequency/inverse document frequency, and report length, and then used these features as input to traditional and ensemble machine learning classifiers. Their experiments showed that logistic regression and multinomial Naive Bayes achieved better accuracy than other methods. Khaloufi [30] proposed a preliminary diagnosis of COVID-19 using symptom monitoring from smartphone embedded sensors. The model achieved an overall accuracy of 79% for detecting the COVID-19 cases. Menni [9] proposed a model combining symptoms to predict COVID-19 cases based on reported symptoms via a smartphone-based app. The study found that loss of smell and taste is a potential predictor of COVID-19 apart other symptoms such as high temperature and a new, persistent cough. Canas [31] presented an early detection model for COVID-19 cases using prospective, observational, longitudinal, and self-reported data from patients in the UK on 19 symptoms over three days after symptom onset. The experimental results showed that the hierarchical Gaussian model achieved higher performance than the logistic regression model.

Some researchers have used FL in the COVID-19 detection system to protect patients’ data because patients’ data are sensitive and impacts patient security. Yan [21] proposed an FL for COVID-19 detection based on chest X-ray images. The study compared performances of four models (MobileNetv2, ResNet18, ResNeXt, and COVID-Net) and found that ResNet18 achieved the highest accuracy in both training with and without FL. Zhang [15] presented a novel dynamic fusion-based FL approach to detect COVID-19 infections using CT and chest X-ray images. The study conducted experiments using the following three models: GhostNet, ResNet50, and ResNet101 and found that the proposed approach achieved better performance than the default setting one for ResNet50 and ResNet101. Abdul [32] presented an FL model to identify COVID-19 cases using chest X-ray images and a descriptive dataset. The study found that using softmax activation function and stochastic gradient descent (SGD) optimizer achieved better performance. A brief summary of existing COVID-19 detection approaches is presented in Table 1.

Unlike existing methods, we presented an FL framework with higher privacy for COVID-19 detection by adding differential privacy stochastic gradient descent (DP-SGD). We also provided strategies for improving model accuracy on Non-IID data and maintaining the robustness between security and accuracy of the COVID-19 detection model. By evaluating the proposed models on two different challenging datasets (chest X-ray images and symptom information), we found that convolutional neural network (CNN) model with adding spatial pyramid pooling (SPP) layer achieved the highest accuracy on the chest X-ray images model and artificial neural networks (ANNs) outperformed the other models such as long short-term memory (LSTM) and 1D CNN (1DCNN) for COVID-19 detection using symptom dataset. To the best of our knowledge, this is the first study to apply the DP-SGD in FL to detect COVID-19 cases based on chest X-ray images and symptoms.

## 3. Approach

In this section, we present a comprehensive overview of our privacy-preserving FL system for COVID-19 detection. This section is organized as follows: first, we introduce the overview of FL in Section 3.1. We then provide a comprehensive introduction of DP-SGD in Section 3.2. We explain the detail of our federated COVID-19 system model in Section 3.3. Finally, Section 3.4 and  Section 3.5 present various network models designed to recognize COVID-19 cases based on chest X-ray images and symptom information, respectively.

### 3.1. Federated Learning

The FL approach was introduced in 2016. It is a machine learning strategy in which multiple clients can collaboratively solve a machine learning problem, with each client storing their own data and sharing or transfering data with other clients [18]. FL ues less storage or computational resources on the central server than centralized learning, and it helps protect each client’s private data.

FL was initially implemented over several small devices [18,33]. The various implemented FL applications have significantly increased, including some, which might involve only a few clients in collaboration among institutions [34,35,36]. These two FL settings are called “cross-device” and “cross-silo”, respectively. A typical FL training is achieved by following several basic steps. In the first step, all chosen clients download the current weight of the master model. Second, the clients compute the weight and update it independently based on their local data. Finally, all clients update their weight to the server, where they are gathered and aggregated to produce a new master model. These steps are repeated until a certain convergence criterion is satisfied.

In our setting, we termed the Federated Averaging (FedAvg) [18] as our FL system. In this manner, the selected clients will compute the gradient of the loss on the current model using their local data for each communication round. Then, the server calculates a weighted average of the resulting models. The pseudo-code of FedAvg adapted from  [18] is given in Algorithm 1.
**Algorithm** **1** FederatedAveraging. The *K* clients are indexed by *k*, *B* is the local minibatch size, *E* is the number of local epochs, and η is the learning rate.**Service executes:**$w_{0}\leftarrow$ random initialization**for** each round *t* = 1,2,... **do**    $S_{t}\leftarrow$ (random subset of max(C×K,1) clients)    **for** each client $k\in S_{t}$ in parallel **do**        $w_{t+1}^{k}\leftarrow$ ClientUpdate(*k*,$w_{t}$)    **end for**    $w_{t+1}\leftarrow \sum_{k=1}^{K}\frac{n_{k}}{n}w_{t+1}^{k}$**end for****Client update**(k,w):Split local dataset in *B* ($\frac{B}{n_{k}}$ batches of size *B*)**for** epoch $e\in \left[1,E\right]$ **do**    **for** batch b∈B **do**        w←w−η▿ℓ(w;b)    **end for****end for**return *w* to server

### 3.2. Differential Privacy Stochastic Gradient Descent (DP-SGD)

Differential privacy is a strong standard for quantifying and limiting personal information disclosure [37,38,39]. It masks the contribution of any individual user by introducing a level of uncertainty into the released model. Privacy loss parameters (ϵ, δ) quantify differential privacy, where ϵ denotes how much a person with output would be able to see the dataset, δ represent the probability that an unwanted event happens that leaks more data than normally. The smaller the (ϵ, δ), the higher the privacy. We have a differential privacy definition as follows: a randomized algorithm *A*: *D*→*R* with domain *D* and range *R* is (ϵ, δ)-differential private if for any subset of outputs S ⊆*R* and for any two adjacent inputs $d,d^{\prime }$∈*D*: $Pr\left[A(d)\in S\right]\leq e^{\epsilon }Pr\left[A(d^{\prime })\in S\right]+\delta$.

In the term of Federated Learning, we say that two decentralized datasets *D* and $D^{\prime }$ are adjacent if they differ in a single entry, that is, if $D^{\prime }$ can be obtained from *D* by adding or subtracting all the records of a single client. δ is preferably smaller than $\frac{1}{|d|}$.

Differential privacy guaranteeing may impact the accuracy or utility of our model. In the context of rich data, it appears that the model can offer both low privacy risk and high utility. However, for large datasets, the optimization methods must be scalable. Therefore, we used SGD to control the influence of training data during the training process as described in previous works [40,41,42] for our differential privacy setting. The DP-SGD strategy adds random Gaussian noise on the aggregated global model that is enough to hide any single client’s update. It consists of the following steps: at each step of the DP-SGD, we compute the gradient for a random subset of examples, then clip these per-sample gradients into a fixed maximum norm $\ell_{2}$. Next, random noise is added to the clipped gradients in computing the average step. Finally, we multiply these clipped and noised gradients using the learning rate and apply the product to update model parameters. Clients perform perturbation on their gradients using the DP-SGD strategy after computing the training gradients based on their local data in Algorithm 1.

### 3.3. System Model

We developed our Federated COVID-19 detection system based on a client-server architecture implementing the FedAvg via local stochastic gradient descent (SGD) and addressing privacy risks with a DP-SGD guarantee. Our federated COVID-19 system includes three stages: clients synchronize with the server, clients compute the local models based on individual data, and the server aggregates the global model. The overall system architecture is shown in Figure 1.

Clients synchronize with the server

At the round *t*, a random C fraction of the clients were selected to connect to the server for computing the gradient of loss over all the data held by these clients. The selected clients download the current average model parameters $\theta_{average}^{t-1}$ from the previous iteration. For the 1st iteration, the clients will use the same random initial model parameters $\theta_{0}$.

Clients compute the local models based on individual data

Each client locally computes the training gradients and updates independently based on their local data divided into *B* mini-batches for *E* epochs. The client then performs perturbation on their gradients using the DP-SGD technique described in Section 3.2. Finally, the client reports the learned model parameters (denoted as $\theta_{k}^{t}$ where *k* is the client index) to the server for averaging.

Server aggregates global model

Once the server receives the parameter updates from clients, the aggregation model will average the updates to produce the new model parameters based on the FedAvg approach.

The overall complexity of the proposed scheme can be expressed as $O\left(t\times K\times E\times B\right)$ at max, where *t*, *K*, *E*, and *B* represent the total number of communication rounds, the total number of clients, the number of local epochs, and the local minibatch-size of clients, respectively.

### 3.4. Network Models Designed for the Recognition of COVID-19 Using Chest X-ray Images

To evaluate the performance of our FL system using chest X-ray images, we used four deep learning models, such as 2D CNN with 5 × 5 convolutional layers (5 × 5 CNN), residual neural networks (ResNets), 2D CNN with 3 × 3 convolutional layers (3 × 3 CNN), and 2D CNN with 3 × 3 convolutional layers and SPP (3 × 3 CNN-SPP). The complexity of each model is $O\left(\omega \right)$, where ω denotes the model parameters. The detailed descriptions are presented in Table 2.

5 × 5 CNN

The 5 × 5 CNN model was used to construct the decentralized classification model for MNIST digit recognition and has shown promising results [18]. The 5 × 5 CNN architecture includes two 5 × 5 convolution layers; the first convolution layer has 32 channels, the second layer has 64 channels, each layer followed with 2 × 2 max pooling, and the fully connected layer with 512 units and ReLu activations. Similar to [18], we used a 5 × 5 CNN with SGD optimizer function for COVID-19 detection using chest X-ray images.

ResNets

ResNets is a specific neural network proposed in [43]. ResNets are made up of residual blocks, which help improve the accuracy of models using skip connections. The skip connections in residual blocks solve the vanishing gradient problem in deep neural networks (DNNs). As a result, the model learns in such a way that the higher layer outperforms the lower layer. The ResNet has multiple variations, namely ResNet16, ResNet18, ResNet34, ResNet50 and ResNet101, which contains 16, 18, 34, 50, and 101 layers, respectively. ResNet has shown a compelling efficiency for COVID-19 detection using chest X-ray data [44,45,46,47]. Similar to these previous works, we applied ResNet18 and ResNet50 as one of the deep learning networks for our COVID-19 detection based on chest X-ray images.

3 × 3 CNN

The 3 × 3 CNN was one of our evaluation models for COVID-19 detection based on chest X-ray images, which includes two 3 × 3 convolutional layers; the first layer with 32 channels, and the second layer with 64 channels followed with 2 × 2 max-pooling layer and a fully connected layer with 128 units, ReLu activation, and a final softmax output layer. Two dropout layers were added before and after the fully connected layer with the dropout probability of 0.25, and 0.5, respectively, to reduce overfitting. Figure 2 shows our proposed 3 × 3 CNN architecture for COVID-19 detection.

3 × 3 CNN-SPP

The SPP layer was first introduced in [48], which helps the CNN model agnostic input image size. The Bag of Words (BoW) approach inspires SPP [49], which pools the features together. SPP outperforms conventional pooling by capturing more spatial information and accepting arbitrary input size. To adopt SPP in a deep network, we must replace the last pooling layer with an SPP layer, such as pooling layer after the last convolutional layer.

A deep network using the SPP layer has shown outstanding accuracy in classification, and detection problems [48,50,51,52,53,54]. In this work, we used an SPP layer with (1 × 2 × 4) spatial bins for our 3 × 3 CNN model before classification. Figure 3 shows our COVID-19 detection using a 3 × 3 CNN with SPP architecture.

### 3.5. Network Models Designed for the Recognition of COVID-19 Using Symptom Data

To evaluate the performance of our FL system using symptom data, we used three machine learning models such as 1DCNN, ANN, and LSTM. The complexity of each model is $O\left(\omega \right)$, where ω denotes the model parameters. The detailed descriptions are presented in Table 3.

1DCNN

CNN not only achieves excellent performance on computer vision tasks such as object detection [55,56], image classification [57], image generation [58], tracking task [59], and face recognition [60], but also can be used to sequence data [61]. Some works have achieved great results in COVID-19 detection using CNN architecture on textual data. For example, Lella et al. [62] used the 1DCNN with augmentation to diagnose COVID-19 in human-based respiratory sounds such as cough, breath, and voice. In this study, we applied the 1DCNN to verify COVID-19 cases based on symptom data. Our model contained three 1DCNN layers, each with a convolutional kernel size of three and a one-step stride. The output size for all layers is 64. After the third layer, the dropout function with a probability of 0.5 is applied to prevent the model from overfitting. Then, a 1D max-pooling layer is applied to reduce the output dimension. Finally, a softmax layer is applied to calculate the probability of each output.

ANN

The ANN is a computer simulation based on the human brain that allows the machine to learn and make decisions [63]. There are different layers in an ANN structure, each layer is arranged as a vector with several single units called neurons. Each input layer performs mathematical processing to produce the output layer, which serves as the input for the next layer. Figure 4 represents the structure of an ANN.

Because of these abilities, ANN has been applied to different machine learning tasks such as time-series prediction [64] and computer vision task [65,66], and has produced reliable results. In the COVID-19 prediction task, several works using the ANN structure have yielded excellent results [67,68,69]. Furthermore, Hayat Khaloufi et at. [30] conducted a highlight study in which a customized ANN was proposed to predict COVID-19 from the collected dataset, which can help predict whether a patient is infected based on their symptoms. The proposed model outperformed other traditional machine learning methods. In this study, we employed an ANN structure with four hidden layers. The hidden size of the four layers is 64, 128, 128, and 2, respectively. A dropout function with a probability of 0.5 is applied after the third layer to reduce the dimensional output vector and speed up the training process. Finally, a softmax function is applied after the last layer to produce the probability of our outputs.

LSTM

An RNN is considered another type of ANN, which can collect data across sequential steps and process one element of sequential data at a time. Unlike the traditional neural networks, the output of RNN depends on the primary elements within the sequence, which makes RNN suitable for solving sequential or time-series data.

The LSTM is a variant of RNN, which was first proposed by Hochreiter et al. [70] to tackle the vanishing and exploding gradients problems that commonly happened from the conventional RNN. In general, a typical LSTM cell is comprised of four different gates such as forget gate, input gate, cell gate, and output gate. The structure of an LSTM cell is presented in Figure 5.

In each operation, the LSTM cell processes an given input sequence $x=\left[X_{1},X_{2},...,X_{T}\right]$ to produce an output hidden sequence $h=\left[h_{1},h_{2},...,h_{T}\right]$ using the following equations iteratively from t=1 to *T*:(1)$$i_{t}=\sigma \left(W_{i}x_{t}+U_{i}h_{t-1}+b_{i}\right),$$
(2)$$f_{t}=\sigma \left(W_{f}x_{t}+U_{f}h_{t-1}+b_{f}\right),$$
(3)$$o_{t}=\sigma \left(W_{o}x_{t}+U_{o}h_{t-1}+b_{o}\right),$$
(4)$${\tilde{c}}_{t}=\tanh\left(W_{c}x_{t}+U_{c}h_{t-1}+b_{c}\right),$$
(5)$$c_{t}=f_{t}⊙c_{t-1}+i_{t}⊙{\tilde{c}}_{t},$$
(6)$$h_{t}=o_{t}⊙\tanh\left({\tilde{c}}_{t}\right),$$
where $f_{t}$, $i_{t}$, ${\tilde{c}}_{t}$, and $o_{t}$ are the forget gate, the input gate, the candidate cell gate, and the output gate at time *t*, respectively; $W_{i}$, $U_{i}$, $W_{f}$, $U_{f}$, $W_{c}$, $U_{c}$, $W_{o}$, $U_{o}$ are the weight matrices; $b_{i}$, $b_{f}$, $b_{c}$, $b_{o}$ are the bias vectors; ⊙ denotes the Hadamard product; σ and tanh denotes a sigmoid and a tangent activation function, respectively.

With the great success in solving sequence data, LSTM has been applied to COVID-19 detection tasks and achieved great performances [71,72,73,74,75]. For instance, ArunKumar et al. [76] proposed a deep learning approach that modified the traditional LSTM with a new activation function for predicting the infected cases and death cases of the COVID-19 dataset. In this paper, we employed a simple stacked of three LSTM layers with a hidden size of 64 to predict our COVID-19 symptom dataset. The LSTM has a dropout layer with a probability of 0.25. After the LSTM, two Linear layers are added to reduce the output dimension. Finally, a softmax layer is applied to calculate the probability of each output.

## 4. Experiment

### 4.1. Data Collection and Processing

Two types of COVID-19 datasets, chest X-ray images and symptom data, were used to train and evaluate our FL system.

#### 4.1.1. Chest X-ray Dataset

Following the work of a group of researchers from Qatar University and the University of Dhaka, Bangladesh, and collaborators from Pakistan, Malaysia, and medical doctors [77,78], we collected a dataset containing 3616 COVID-19 positives, 10,192 normal and 1345 viral pneumonia chest X-ray images. The COVID-19 data were collected from the various publicly accessible datasets, online sources, and published papers [79,80,81,82,83,84], normal data were collected from two different datasets [85,86], and viral pneumonia data were collected from chest X-ray images (pneumonia) database [86]. Few samples of chest X-ray images are shown in Figure 6.

For each class, we randomly kept 200 images for testing data and the rest for training. The statistics of the chest X-ray dataset are shown in Table 4.

#### 4.1.2. Symptom Dataset

The symptom dataset for COVID-19 cases is based on a list of symptoms published by WHO in May 2020 from India. It is provided by [87]. The symptom dataset contains 5434 samples with 21 columns such as breathing problem, fever, dry cough, sore throat, running nose, asthma, chronic lung disease, headache, heart disease, diabetes, hypertension, fatigue, gastrointestinal, abroad travel, contact with COVID-19 patient, attended a large gathering, visited public exposed places, family working in public exposed places, wearing masks, sanitization from markets, and COVID-19. Each column contains a “Yes” or “No” value.

We did some data processing before feeding the data into the network model using the following steps:Removing the columns containing unique values because these columns provide no useful information for our model;Converting categorical data to one-hot encoding data as follows: 1 represents “Yes” and 0 represents “No”;For each COVID-19 class, we randomly kept 10% for testing and used the rest (90%) for training.

Table 5 presents the statistics of the symptom dataset.

### 4.2. Improvement in COVID-19 Detection Based on Chest X-ray Images and 3 × 3 CNN-SPP

To choose an effective model for our federated COVID-19 detection based on chest X-ray images, we conducted experiments using four different models discussed in Section 3.4. We tested each model by running experiments with the number of clients K = 3, client-fraction C = 0.33 (one client per round), local epoch E = 1, client learning rate η = 0.02, and the local minibatch size B = 20. As shown in Figure 7, the 3 × 3 CNN model achieved better accuracy than 5 × 5 CNN, Resnet18, and Resnet50 methods. Furthermore, the model was further improved by adding an SPP layer. Our proposed 3 × 3 CNN-SPP model achieves the highest accuracy of 95.32% after 1000 communication rounds. Adopting the SPP layer allows our deep convolutional neural network is able to generate representations from arbitrarily sized images. The 3 × 3 CNN-SPP is therefore able to extract features at different scales and capture more spatial information. As such, the classification performance is improved. Therefore, we used this 3 × 3 CNN-SPP model for our COVID-19 detection system based on chest X-ray images. The 5 × 5 CNN model achieved a faster convergence compared to the other models. In addition, the accuracy of all models steadily increases after 200 communication rounds and keeps being stable.

### 4.3. Improvement in COVID-19 Detection Based on Symptom Data and ANN

We evaluated three models discussed in Section 4.3 for our federated COVID-19 detection using symptom data. For each model, we conducted the experiment with the number of clients K = 4, client-fraction C = 0.25 (one client per round), local epoch E = 1, client learning rate η = 0.02, and local minibatch size B = 20. As shown in Figure 8, our proposed ANN achieved a more favorable performance than 1DCNN and LSTM, with the highest accuracy of 96.65%. Owning the functionality of hosting multiple data points through the neurons to perform the mathematical processing to produce the outputs. ANN applies the learnable weight for each neuron and updates it by the cost function after each iteration to fit the training data. For that reason, the ANN has shown great success when dealing with a dataset that has a non-linear between the input and output variables. As such, the accuracy is improved. Therefore, we used this ANN model as our COVID-19 detection model using the symptom dataset. The accuracy of 1DCNN and ANN models steadily increases after 200 communication rounds and keeps stable until reaching a certain convergence at 600 communication rounds, while the LSTM model reaches the convergence very early at the first round.

### 4.4. IID and Non-IID

Unlike centralized models, FL usually faces a Non-IID problem [18] in which the size and distribution of local datasets will typically vary heavily between different clients because each client corresponds to a particular user, geographic location, or time window. For instance, the chest X-ray dataset with an imbalance label size is shown in Figure 9, where each client owns data samples with a fixed number of k label classes. It resulted in the client’s local models having the same initial parameters converging to different models, and aggregating the divergent models on the server can slow down convergence and worsen the learning performance.

In this study, we compared the performance of our model on both IID and Non-IID datasets. For the IID dataset, each client is randomly assigned a uniform distribution over all classes. For the Non-IID dataset, we first sort the data using the class label. We then divide data into two cases: (1) Non-IID(1), 1-class non-IID, where each client receives data partition from only a single class, and (2) Non-IID(2), 2-class Non-IID, where each client receives data partition from not more than two classes. We did not consider Non-IID(2) for the symptom dataset because the symptom dataset only contains two classes.

We used the same parameters in Section 4.2 and Section 4.3 for chest X-ray and symptom models. As shown in Figure 10 and Table 6, a significant reduction is observed on Non-IID data than IID data on our chest X-ray images. The maximum accuracy reduction occurs for the most extreme 1-class non-IID(1), approximately 49.71 to 55.32%. For 2-class non-IID(2), the accuracy reduction was approximately 14 to 24%.

A similar observation was made with the symptom dataset; Figure 11 and Table 7 show a 1.28% to 2.29% reduction in accuracy in Non-IID(1) compared to IID.

From these experiments, we found that non-IID data are one of the major issues of the FL system because non-IID contains different sizes and distributions of local datasets for each client. It resulted in the client’s local models being significantly different from each other, and aggregating the divergent models on the server can slow down convergence and significantly reduce model accuracy. Therefore, we must find a way that can improve our model performance on non-IID data. In the next section, we will propose a strategy to improve our Non-IID(1) data performance.

### 4.5. Non-IID Improvement

In this section, we evaluate different parameters in our Non-IID(1) setting to determine the relationship between these parameters and our model performance. We first experiment with total client K, client-fraction C, and local mini batch-size B on models using chest X-ray images. We then further validate these parameters on the model using the symptom dataset.

#### 4.5.1. Non-IID with Different Numbers of Client

We first evaluated the model’s performance using chest X-ray images on Non-IID(1) with various total numbers of clients (3, 30, and 300) while keeping the other parameters: client-fraction C = 0.33, local epoch E = 1, client learning rate η = 0.02, and the local minibatch size B = 20. Figure 12 and Table 8 show the impact of varying K for our COVID-19 detection model. The results demonstrated that using a larger number of clients (K = 30 and K = 300) significantly improves for our Non-IID(1) setting than the model using K = 3. This can be explained by using many clients for our Non-IID(1) setting; some clients may receive data from the same class. Therefore, if our model has previously learned similar data patterns from previous clients, it will easily recognize the patterns of the current client. This improved the model’s accuracy on chest X-ray images.

#### 4.5.2. Increasing Client-Fraction

In this experiment, we evaluated our model using chest X-ray images with client-fraction C, which controls the amount of multi-client parallelism. To compute this, we fixed local epoch E = 1, batch-size B = 20, client learning rate η = 0.02, and number of total client K = 300 (achieved best performance in previous Section 4.5.1), while changing the ratio of client-fraction C with varying value ∈ {0.1, 0.3, 0.6, 0.9, and 1.0}. As shown in Figure 13, using the larger client-fraction improved our model accuracy. Moreover, the model with a larger client-fraction helped the model converge faster with the same number of communication rounds.

#### 4.5.3. Increasing Computation per Client

The local batch-size B is the last parameter we used to evaluate the effect on the Non-IID(1) model. We fixed client-fraction C to 1.0, which showed the improvement results in the previous section, local epoch E to 1, client learning rate η to 0.02, and the number of clients K to 300, while the local batch-size value will be selected via varying value ∈ {1, 20, 100, 200, and 500}. Figure 14 shows that with 1000 communication rounds, the model using chest X-ray images with small batch-size B = 1 achieved the lowest accuracy of approximately 63.33%, while larger batch-size B = 20 achieved an improvement result with 77.56% of accuracy, and the best result was achieved by the model using largest batch-size B = 200 with 79.56% of accuracy. In addition, the model using a larger batch-size achieves convergence faster compared to the model with a smaller batch-size because a larger batch-size means using a larger amount of data.

#### 4.5.4. NonIID and Refined NonIID

In Section 4.5.1, Section 4.5.2 and Section 4.5.3, we discovered that each component has affection on chest X-ray model performance. We then combined these parameter values to see how much our model accuracy can be increased compared to the baseline model. The baseline model is Non-IID(1) used in Section 4.4 with the number of clients K = 3, fraction C = 0.33, client learning rate η = 0.02, and local batch-size B = 20. Non-IID(300 clients) is a modification of the baseline model with K = 300 clients. The final refined model is a refined model from Non-IID (300 clients) with the number of client-fraction = 1.0, and local batch-size = 200.

Table 9 shows that using K = 300 clients helped improve accuracy by up to 33.97% compared to the baseline model using three clients. Furthermore, we further improved the accuracy model of Non-IID (300 clients) by up to 4.7% with an increasing client-fraction and local batch-size.

#### 4.5.5. Experiments on Symptom Data

From Section 4.4, we observed a slight deduction of the symptom model on Non-IID(1). We now examined whether tuning parameters (number of total clients K, client-fraction C, and local mini batch-size) of the chest X-ray model are also efficient with the symptom model.

As shown in Figure 15, using a larger client-fraction C showed slight improvement for model performance on Non-IID(1). However, increasing the number of clients K and batch-size B did not improve the performance of our symptom model on Non-IID(1), as shown in Figure 16.

Table 10 shows that our refined model has slightly improved with 0.31 to 0.82% accuracy compared to Non-IID(1) and almost reached the model’s accuracy on IID data.

### 4.6. Privacy Improvement for Federated COVID-19 Detection Model Using DP-SGD

FL helps mitigate privacy risks associated with centralized machine learning without sharing each client’s private data. However, the adversary might infer our information from the shared gradients from the previous model of our FL system. To make it much more challenging for an adversary to breach privacy, we applied DP-SGD by adding a random Gaussian noise on local gradients in the aggregating step on the server’s model. In this section, we first examine how much the different levels of privacy affects our model performance. We then provide the strategy to keep the robustness of differential privacy and model’s accuracy. Our FL with DP-SGD for a COVID-19 detection system is shown in Figure 1.

#### 4.6.1. Trade-Off between the Model Privacy and Accuracy

For the chest X-ray model, we conducted experiments for the IID dataset setting with number of total clients K = 3 and with varying noise values to see how much differential privacy impacts our utility model. A lower ϵ value means the model has higher security. We set δ = $10^{-5}$, client-fraction = 0.33, client learning rate η = 0.02, and batch-size = 20. As shown in Figure 17, we found a trade-off between our privacy and model accuracy. The more noise we add to our model, the more reduced our model accuracy is.

The similar observations were made when experiments on the symptom model. Figure 18 presents a trade-off between model privacy and performance when we conducted experiments with varying noise values. When the noise value was set to 1.0, the model achieved the highest accuracy, but it was less secure with the largest ϵ value = 1600. In contrast, the model achieved the most security but also worse accuracy when the noise value was set to 5.0. We kept the number of total clients K to 4, client-fraction C to 0.25, client learning rate η to 0.02, δ = $10^{-5}$, and batch-size to 20 in these experiments.

#### 4.6.2. Robustness of Differential Privacy

Section 4.6.1 demonstrated the trade-off between privacy and the model’s accuracy. Therefore, we wanted to find a way that helps us reduce the model privacy risk, but we can still keep the similar utility of our model. We first experimented with three system parameters: fraction of model q (number of clients per round/total number of clients), number of total clients, and the noise value on our chest X-ray model. In our experiments, we scaled up the total number of clients while keeping the fraction of the model constant, and the noise value was scaled up using varying values ∈ {0.1, 0.3, 0.7, 1.3, 1.9, and 2.1}. As shown in Figure 19 and Table 11, by increasing the total number of clients while keeping the fraction of the model constant, we could add the larger noise value, and the model achieved a similar utility and higher privacy. For example, after increasing the number of clients from 180 to 270, we could increase the noise value from 1.3 to 1.9, lowering a half ϵ value from 97.39 to 46.36 while the model’s accuracy only reduced by 0.94% from 70.21% to 69.27%.

The same conclusion was reached on the symptom model; we achieved robustness of differential privacy and accuracy by increasing the total number of clients while keeping the fraction of model constant, and scaling up noise proportionally. As shown in Table 12, after increasing the number of clients from four to sixty while keeping the fraction of model q constant, we could reduce by more than ten times ϵ value from 1600 to 120 while the model’s accuracy only reduced by 0.17% from 93.56% to 93.39%.

## 5. Conclusions

This study presented a higher privacy-preserving FL system for COVID-19 detection based on two types of datasets: chest X-ray images and symptom data. Through experiments on seven deep learning models: 5 × 5 CNN, ResNets, 3 × 3 CNN, 3 × 3 CNN-SPP, ANN, 1DCNN, and LSTM, our federated COVID-19 detection model using 3 × 3 CNN-SPP and ANN achieved the best accuracy of 95.32% on chest X-ray images and 96.65% on symptom data, respectively. We first showed that the accuracy of FL for COVID-19 identification reduced significantly for Non-IID data. As a solution, we proposed a strategy to improve accuracy on Non-IID data by increasing the total number of clients, parallelism (client–fraction), and computation per client (batch size). Experiments showed that Non-IID model accuracy could be increased by 18.41% for chest X-ray images and 0.82% for symptom data. Second, to enhance patient data privacy for our FL model, we applied a DP-SGD that is resilient to adaptive attacks using auxiliary information. Finally, we proposed a strategy to keep the robustness of FL to ensure the security and accuracy of the model by keeping the fraction of the model constant and proportionally scaling up the total number of clients and the noise.

In our future work, we would like to implement this method using a larger dataset available from various hospitals worldwide. Furthermore, we hope that our proposed privacy-preserving FL framework enhances data protection for collaborative research to fight the COVID-19 pandemic.

## Figures and Tables

**Figure 1 sensors-22-03728-f001:**
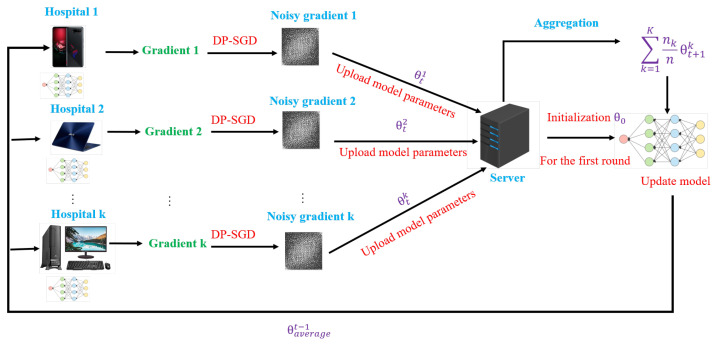
Federated COVID-19 detection system architecture.

**Figure 2 sensors-22-03728-f002:**
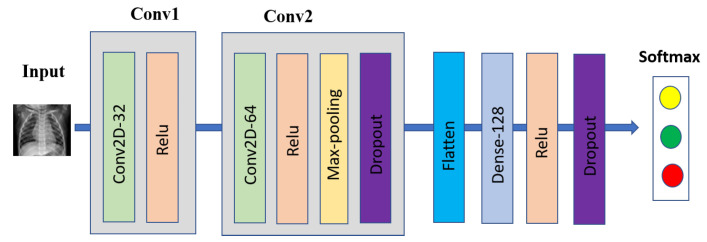
The 2D CNN with 3 × 3 convolutional layers architecture.

**Figure 3 sensors-22-03728-f003:**
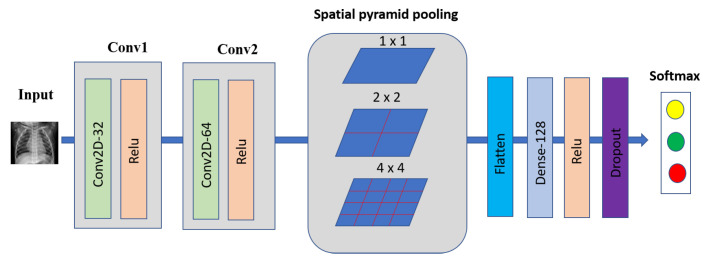
The 2D CNN with 3 × 3 convolutional layers and SPP architecture.

**Figure 4 sensors-22-03728-f004:**
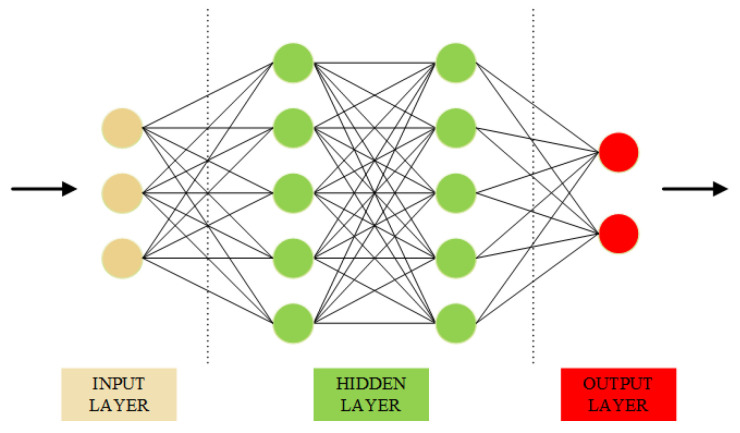
The structure of ANN.

**Figure 5 sensors-22-03728-f005:**
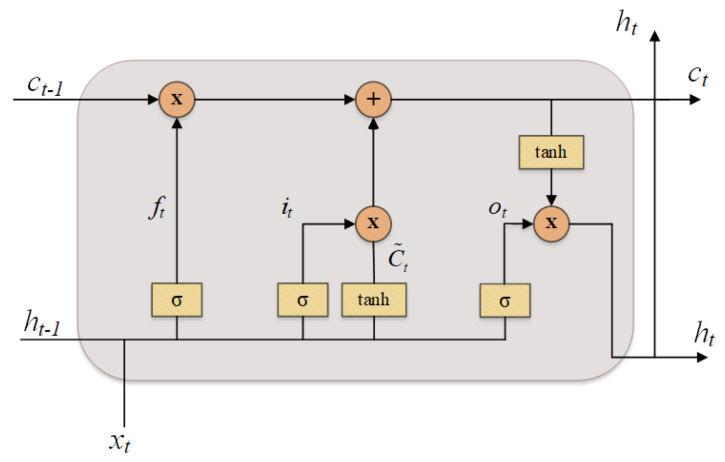
The structure of one cell conventional LSTM.

**Figure 6 sensors-22-03728-f006:**
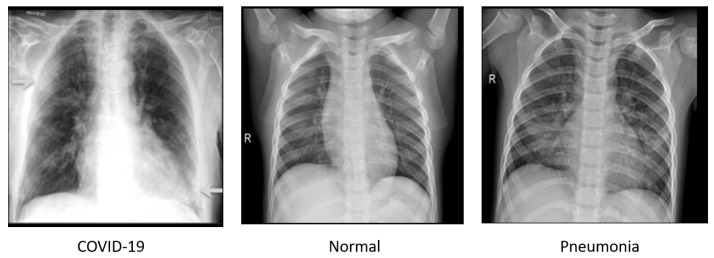
Few samples of chest X-ray images.

**Figure 7 sensors-22-03728-f007:**
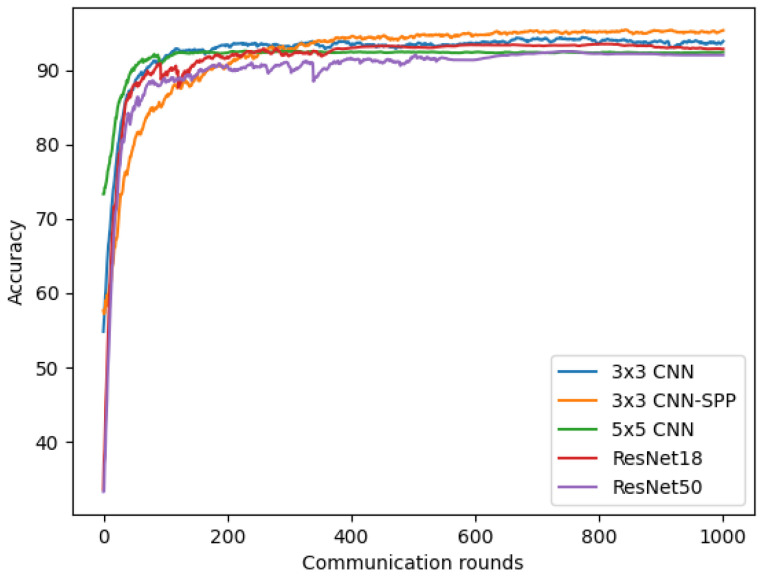
Improvement in COVID-19 detection based on chest X-ray images and 3 × 3 CNN-SPP.

**Figure 8 sensors-22-03728-f008:**
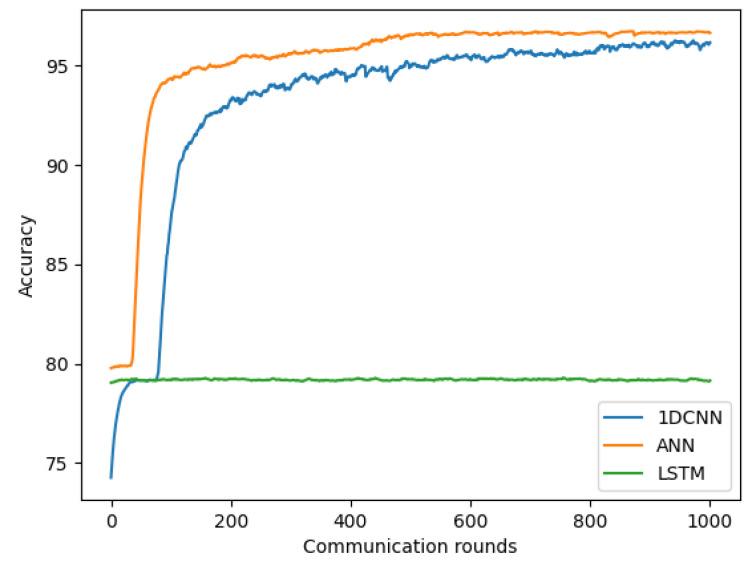
Improvement in COVID-19 detection based on symptom data and ANN.

**Figure 9 sensors-22-03728-f009:**
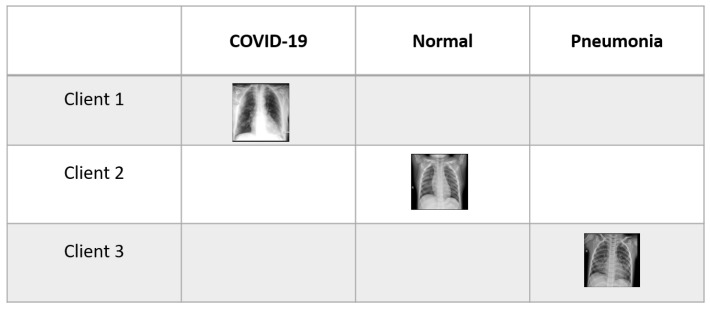
An illustrative example of imbalance size for chest X-ray data with three clients and k = 1.

**Figure 10 sensors-22-03728-f010:**
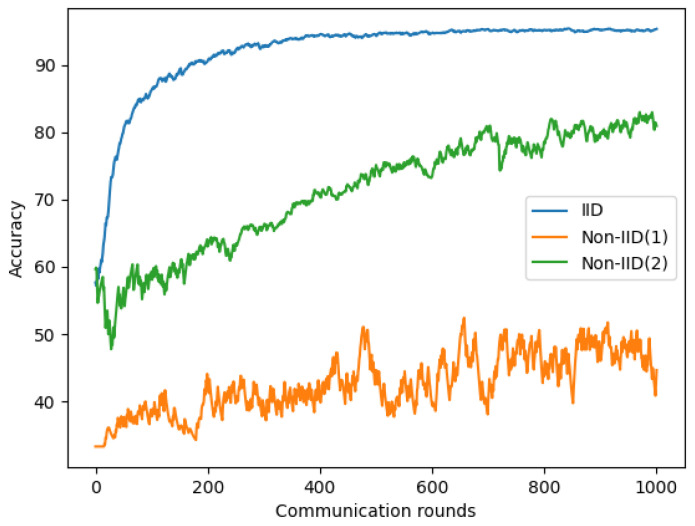
Performance comparison between IID and Non-IID based on chest X-ray images.

**Figure 11 sensors-22-03728-f011:**
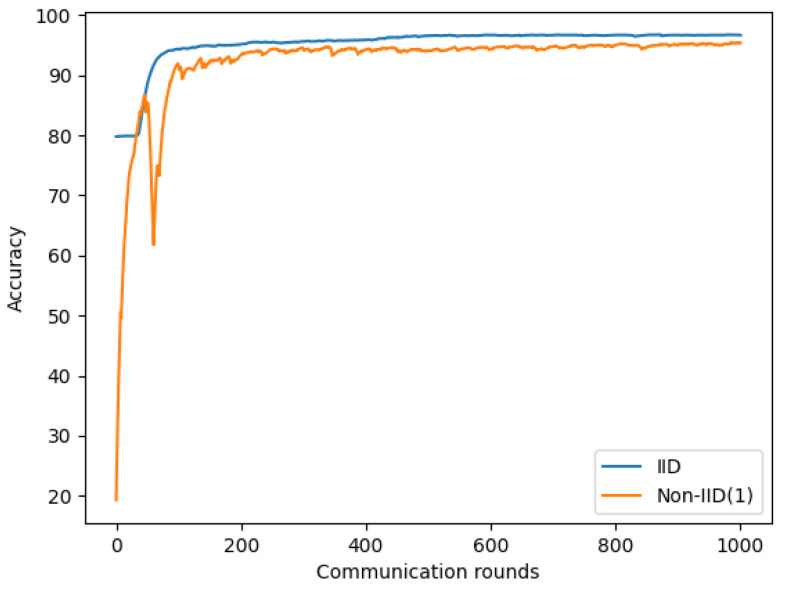
Performance comparison between IID and Non-IID based on symptom dataset.

**Figure 12 sensors-22-03728-f012:**
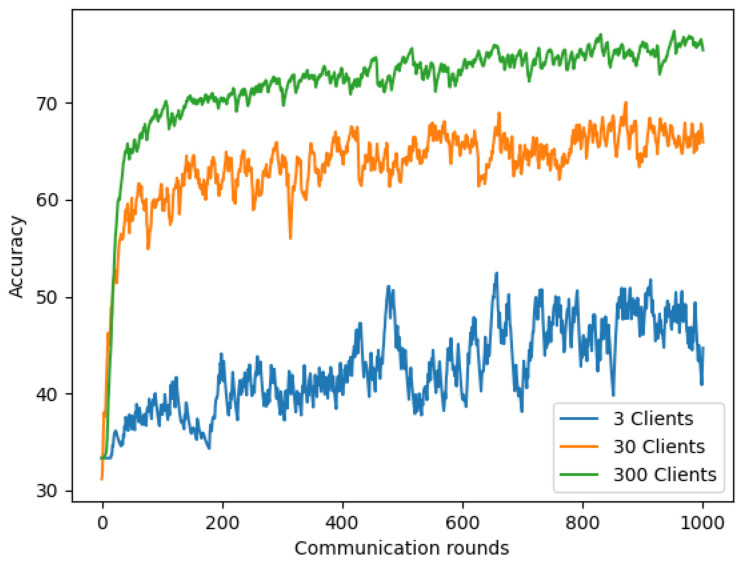
Non-IID with different numbers of clients based on chest X-ray images.

**Figure 13 sensors-22-03728-f013:**
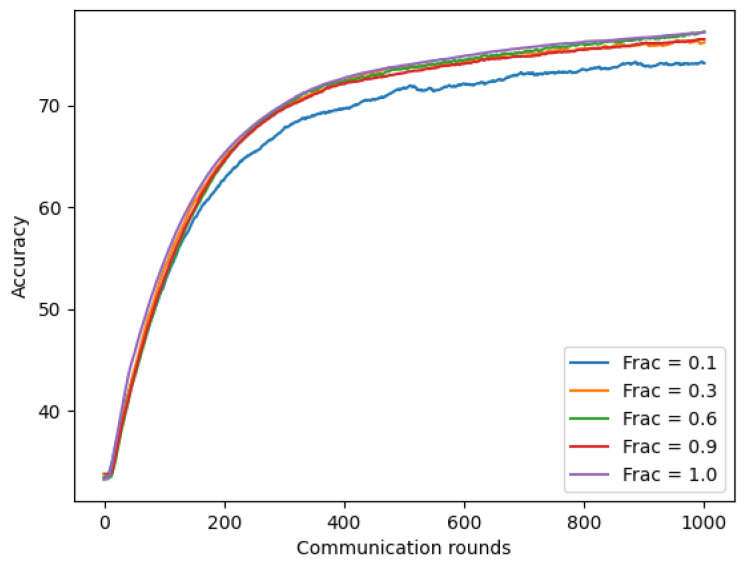
Non-IID with different client-fraction based on chest X-ray images.

**Figure 14 sensors-22-03728-f014:**
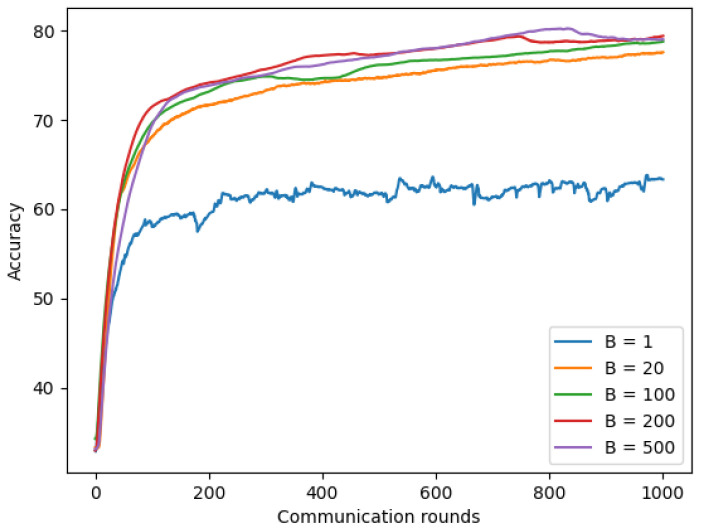
Non-IID with different batch-size based on chest X-ray images.

**Figure 15 sensors-22-03728-f015:**
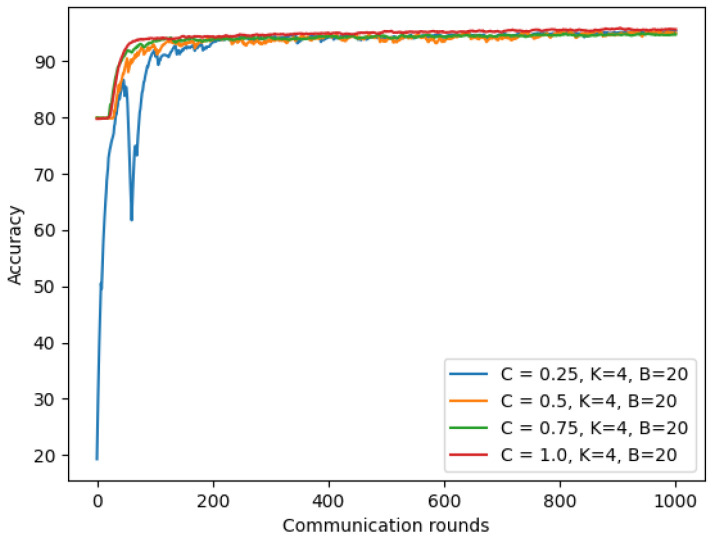
Non-IID with different client-fraction based on symptom data.

**Figure 16 sensors-22-03728-f016:**
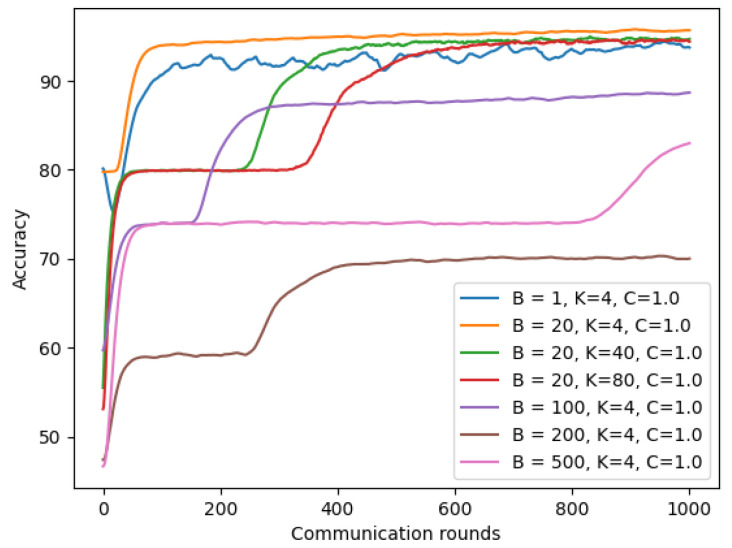
Non-IID with different batch-size and client based on symptom data.

**Figure 17 sensors-22-03728-f017:**
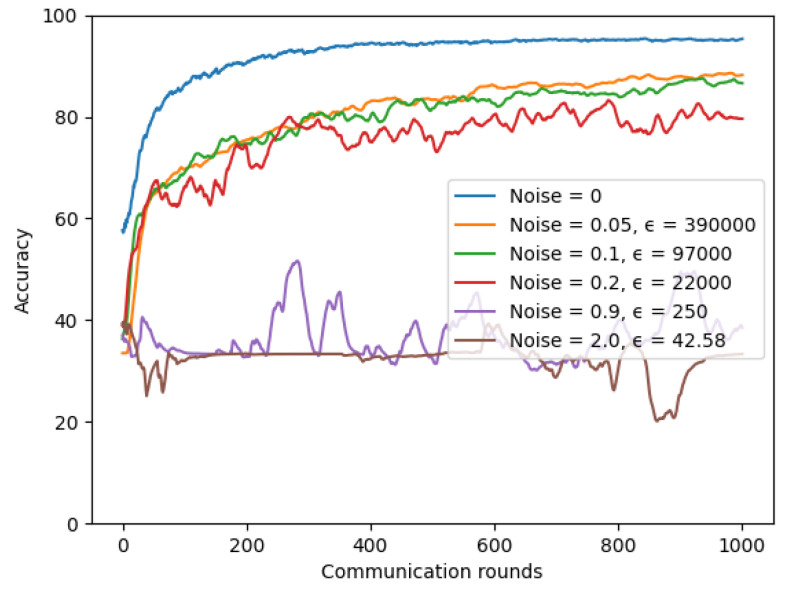
Differential privacy with different noise values on chest X-ray model.

**Figure 18 sensors-22-03728-f018:**
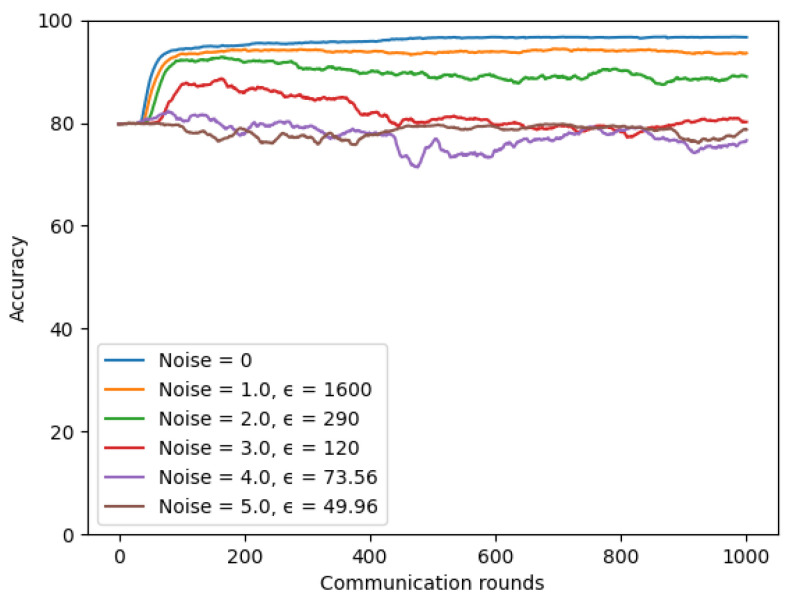
Differential privacy with different noise values on symptom model.

**Figure 19 sensors-22-03728-f019:**
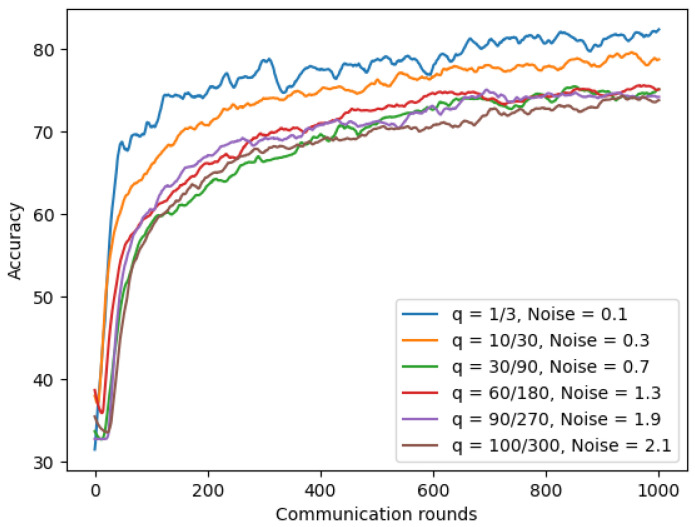
The robustness of differential privacy on chest X-ray images.

**Table 1 sensors-22-03728-t001:** The summary table of existing COVID-19 detection approaches.

Author	Title	Data	Approach	Limitations
Horry et al. [14] (2020)	COVID-19 detection through transfer learning using multimodal imaging data	X-ray, ultrasound, CT scan	VGG19	Sharing sensitive data of patients
Afshar et al. [26] (2020)	Covid-caps: A capsule network-based framework for identification of COVID-19 cases from X-ray images	Chest X-ray	COVID-CAPS	Sharing sensitive data of patients
Mukherjee et al. [27] (2021)	Deep neural network to detect COVID-19: one architecture for both CT Scans and Chest X-rays	Chest X-ray, CXR images	CNN tailored Deep Neural Network	Sharing sensitive data of patients
Otoom et al. [28] (2020)	An IoT-based framework for early identification and monitoring of COVID-19 cases	COVID-19 symptom	Eight algorithms (SVM, neural network, Naïve Bayes, KNN, decision table, decision stump, OneR, ZeroR)	Sharing sensitive data of patients
Akib et al. [29] (2020)	Machine learning based approaches for detecting COVID-19 using clinical text data	COVID-19 symptom	Logistic regression, multinomial Naïve Bayes	Sharing sensitive data of patients
Khaloufi et al. [30] (2021)	Deep Learning Based Early Detection Framework for Preliminary Diagnosis of COVID-19 via Onboard Smartphone Sensors	COVID-19 symptom	ANN, AI-enabled framework to diagnose COVID-19 using a smartphone	Sharing sensitive data of patients
Menni et al. [9] (2020)	Real-time tracking of self-reported symptoms to predict potential COVID-19	COVID-19 symptom	Smartphone-based app, logistic regressions	Sharing sensitive data of patients
Canas et al. [31] (2021)	Early detection of COVID-19 in the UK using self-reported symptoms: a large-scale, prospective, epidemiological surveillance study	Chest X-ray images	MobileNetv2, ResNet18, ResNeXt, COVID-Net	Sensitive data can still be revealed through model updates
Zhang et al. [15] (2021)	Dynamic fusion-based federated learning for COVID-19 detection	Chest X-ray images, CT scan	GhostNet, ResNet50, ResNet101	Sensitive data can still be revealed through model updates
Abdul et al. [32] (2021)	COVID-19 detection using federated machine learning	Chest X-ray, descriptive dataset	Federated with SGD optimizer	Sensitive data can still be revealed through model updates

**Table 2 sensors-22-03728-t002:** The complexity of the proposed models using chest X-ray images.

Model	No. Layers	No. Parameters ω (Milion)
5 × 5 CNN	3	22
ResNet18	18	11.2
ResNet50	50	23.5
3 × 3 CNN	3	1.6
3 × 3 CNN-SPP	4	0.2

**Table 3 sensors-22-03728-t003:** The complexity of the proposed models using symptom data.

Model	No. Layers	No. Parameters ω (Thousand)
1DCNN	5	37.4
ANN	4	26.3
LSTM	5	90.2

**Table 4 sensors-22-03728-t004:** The statistics of the chest X-ray dataset.

	Covid	Normal	Viral Pneumonia	Total Images
Training	3416 images	9992 images	1145 images	14,553 images
Testing	200 images	200 images	200 images	600 images

**Table 5 sensors-22-03728-t005:** The statistics of the symptom dataset.

	Covid	Non-Covid	Total
Training	3949 images	941 images	4890 images
Testing	434 images	110 images	544 images

**Table 6 sensors-22-03728-t006:** The summary table of the performance comparison between IID and Non-IID based on chest X-ray images.

Round	IID	Non-IID(1)	Non-IID(2)
400	**94.50%**	40.56%	70.38%
600	**94.72%**	39.40%	73.45%
800	**95.17%**	45.46%	78.62%
1000	**95.32%**	44.68%	80.93%

**Table 7 sensors-22-03728-t007:** The summary table of the performance comparison between IID and Non-IID based on symptom dataset.

Round	IID	Non-IID(1)
400	**95.88%**	94.24%
600	**96.68%**	94.39%
800	**96.67%**	95.03%
1000	**96.65%**	95.37%

**Table 8 sensors-22-03728-t008:** The summary table of the accuracy of Non-IID with different numbers of clients based on chest X-ray images.

Round	3 Clients	30 Clients	300 Clients
400	40.56%	64.57%	**73.36%**
600	39.40%	65.93%	**73.37%**
800	45.46%	66.52%	**74.89%**
1000	44.68%	65.93%	**75.42%**

**Table 9 sensors-22-03728-t009:** Non-IID with refined Non-IID based on chest X-ray images.

Round	Baseline	Non-IID (300 Clients)	Refined Non-IID
400	40.56%	73.36%	**77.30%**
600	39.40%	73.37%	**78.07%**
800	45.46%	74.89%	**78.59%**
1000	44.68%	75.42%	**79.56%**

**Table 10 sensors-22-03728-t010:** Non-IID with refined Non-IID based on symptom data.

Round	IID	Non-IID(1)	Refined Non-IID(1)
400	95.88%	94.24%	95.06%
600	96.68%	94.39%	95.17%
800	96.67%	95.03%	95.52%
1000	96.65%	95.37%	95.68%

**Table 11 sensors-22-03728-t011:** The summary table of the robustness of differential privacy on chest X-ray images.

q	Noise	Accuracy	ϵ
1/3	0.1	**86.58%**	9.7 × $10^{4}$
10/30	0.3	75.36%	8.9 × $10^{3}$
30/90	0.7	68.35%	5.6 × $10^{2}$
60/180	1.3	70.21%	97.39
90/270	1.9	69.27%	46.36
100/300	2.1	68.70%	**39.40**

**Table 12 sensors-22-03728-t012:** The robustness of differential privacy on symptom data.

q	Noise	Accuracy	ϵ
1/4	1.0	93.56%	1.6 × $10^{3}$
10/40	2.0	**93.99%**	2.9 × $10^{2}$
15/60	3.0	93.39%	1.2 × $10^{2}$
20/80	4.0	91.33%	73.56
25/100	5.0	88.73%	**49.96**

## Data Availability

Public data can be freely accessed and downloaded at https://figshare.com/articles/dataset/COVID-19_Image_Repository/12275009 (accessed on 12 June 2021), https://sirm.org/category/senza-categoria/COVID-19/ (accessed on 12 June 2021), https://github.com/ieee8023/covid-chestxray-dataset (accessed on 12 June 2021), https://github.com/armiro/COVID-CXNet (accessed on 12 June 2021), https://www.kaggle.com/c/rsna-pneumonia-detection-challenge/data (accessed on 12 June 2021), https://www.kaggle.com/paultimothymooney/chestxray-pneumonia (accessed on 12 June 2021), https://www.kaggle.com/hemanthhari/symptoms-and-covid-presence (accessed on 12 December 2021).

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
