# Peer review of "FedSGDCOVID: Federated SGD COVID-19 Detection under Local Differential Privacy Using Chest X-ray Images and Symptom Information"

_sensors, 2022, doi:10.3390/s22103728_

Round 1
Reviewer 1 Report
The authors have provided an approach to increase the privacy of FL in COVID-19 samples. The research motivation and background are clearly presented with necessary citations. However, a few minor modifications are proposed to improve the quality of the manuscript.
1) In line 182, the authors have mentioned the pseudocode (Algorithm 1). It is not clear whether this is a contribution/novel algorithm. If so please highlight it.
2) The DP-SGD method is explained in Fig. 1. Is there any relationship between this methodology and the pseudocode (Algorithm 1) explained in page 5. If so, please highlight this.
3) In Line 344, the authors have mentioned that 3x3CNN SPP models achieves the highest accuracy. Any insights into why this is the case?
4) Similarly, in line 353, the authors have selected ANN? Any insight into why ANN provided better results?
5) In line 356, can you please explain and reference the NON-IID problem
Reviewer 2 Report
The authors proposed a privacy-preserving federated learning scheme for COVID-19 detection without revealing data among owners. The idea is very interesting. I have the following concerns.
1. Please revise the grammatical errors.
2. The very similar technology is considered in this literature ""FBI: A Federated Learning-Based Blockchain-Embedded Data Accumulation Scheme Using Drones for Internet of Things," in IEEE Wireless Communications Letters". Please make a comparison with it.
3. Please discuss the complexity of the proposed scheme.
4. The convergence of the algorithms should also need to be discussed.
5. Add a summary table for Section II along with their limitations to clear the contribution.
6. LSTM demands sequential or time-series data. Will it be a valid comparison with other models?
7. Technical depth is missing. The authors discussed more on the proposed scheme.
8. The training is conducted on very little data. It would be better if more data can be added to the experiment.
9. The authors considered FedAvg in the aggregation. However, FedAVG considers the sample count in the local dataset (n_k in Algorithm 1) and the total sample from all the participants (n in Algorithm 1) during the aggregation. How aggregator will know this? Won't it violate the privacy of the data owner? If the user sent it, how aggregator will know that this is a valid sample number?
Round 2
Reviewer 2 Report
I am recommending to accept this paper.